# Photo-Electro-Thermal Model and Fuzzy Adaptive PID Control for UV LEDs in Charge Management

**DOI:** 10.3390/s23135946

**Published:** 2023-06-27

**Authors:** Yuhua Wang, Tao Yu, Zhi Wang, Yang Liu

**Affiliations:** 1Changchun Institute of Optics, Fine Mechanics and Physics, Chinese Academy of Sciences, Changchun 130033, China; wangyuhua21@mails.ucas.ac.cn (Y.W.); liuyang21h@mails.ucas.ac.cn (Y.L.); 2University of Chinese Academy of Sciences, Beijing 100049, China; 3School of Electronic Information Engineering, Changchun University of Science and Technology, Changchun 130022, China; 4School of Fundamental Physics and Mathematical Sciences, Hangzhou Institute for Advanced Study, University of Chinese Academy of Sciences, Hangzhou 310024, China

**Keywords:** inertial sensor, charge management system, UV LED, photo-electro-thermal model, fuzzy adaptive PID control

## Abstract

Inertial sensors can serve as inertial references for space missions and require charge management systems to maintain their on-orbit performance. To achieve non-contact charge management through UV discharge, effective control strategies are necessary to improve the optical power output performances of UV light sources while accurately modeling their operating characteristics. This paper proposes a low-power photo-electro-thermal model for widely used AlGaN-based UV LEDs, which comprehensively considers the interaction of optical, electrical, and thermal characteristics of UV LEDs during low-power operations. Based on this model, an optical power control system utilizing a fuzzy adaptive PID controller is constructed, in which a switch is introduced to coordinate the working state of the controller. Thus, the steady-state performance is effectively improved while ensuring dynamic performance. The results show that the proposed model has an average prediction error of 5.8 nW during steady-state operations, and the fuzzy adaptive PID controller with a switch can reduce the fluctuation of light output to 0.67 nW during a single discharge task, meeting the charge management requirements of high-precision inertial sensors.

## 1. Introduction

Inertial sensors are crucial instruments for providing inertial references in space gravity missions [1]. However, the accumulation of charges caused by cosmic rays and solar particles affects their low-frequency sensitivity [2]. Therefore, it is necessary to implement a special charge management system (CMS) for on-orbit discharge operations [3,4]. Ultraviolet (UV) discharge technology based on the photoelectric effect has been recognized as a suitable non-contact charge management method for precision gravitational tasks, such as space gravitational wave detection due to its low-disruptive characteristics [5]. It uses controllable UV beams to illuminate the surfaces of inertial sensors, exciting photoelectrons and controlling their flow through a local electric field, thereby removing residual charges without contact [6]. The overall effect is equivalent to connecting an invisible wire to maintain an equal potential state [7].

Due to the complexity of the space environment, CMSs require high reliability, low power consumption, and convenient driving light sources [8]; AlGaN-based UV light emitting diodes (LEDs) meet these requirements. Therefore, in recent years, extensive research has been carried out on their space environment adaptation [9,10], lifetime [11], optical coupling methods [12], spectral stability [13], and charge management technologies based on these advancements [14,15,16]. However, the accurate modeling of interdependent operating characteristics, as well as the suppression of noise interference and model uncertainty in the optical output link through control methods, are often neglected in these studies. In fact, these aspects are closely related to achieving a more accurate discharge, and this paper will investigate them.

For modeling the complex operating characteristics of UV LEDs, the traditional linear model that only considers the simple relationship between current and optical power cannot achieve satisfactory results in high-precision applications. To address this, Hui S and Tao X [17,18] proposed steady-state and dynamic photo-electro-thermal (PET) models for general LED systems in 2009 and 2012, respectively. These models comprehensively consider the interdependent characteristics of actual LED systems, providing valuable theoretical support for their design and analysis. However, the original models involve several variables that are difficult to obtain, making them inconvenient for practical applications [19]. To overcome this, we aim to develop a more convenient and practical low-power PET (L_PET) model for UV LEDs based on their practical characteristics during low-power operations in CMSs.

In the field of optical power control for UV LEDs, a fuzzy adaptive PID (FA_PID) control, which has both the robustness and adaptability of fuzzy control and the high precision of a PID control, is an effective method due to its ability to dynamically adjust control parameters based on real-time feedback [20]; it is widely used in various LED lighting applications [21,22]. However, the addition of the fuzzy inference process also affects its dynamic performance, making its response slower. In this paper, an optical power control system based on an FA_PID controller was constructed to ensure fast response and stable output in CMSs. By introducing a switch to adjust the working state of the controller at different stages, the comprehensive optimization of the dynamic performance and steady-state performance of UV LED light output is realized while maintaining stability and reliability in the presence of uncertainties caused by thermal and attenuation effects.

The aforementioned L_PET model and optical power control based on the FA_PID controller with a switch will help deepen our understanding of the operational characteristics of UV LEDs and improve their performance in practical applications. These efforts will not only contribute to the future design and implementation of a truly feasible and effective non-contact CMS for high-precision inertial sensors, but also provide directions for UV LEDs in light curing, disinfection, decontamination, non-line-of-sight communication applications, etc., in low-power situations.

## 2. UV LEDs in Charge Management

The high-precision inertial sensor used for space gravitational wave detection requires the residual charges on the test mass to always be below $3\times 10^{-12}$ C [23,24]. However, the discharge process that removes charges also generates noise. Thus, the discharge rate must be balanced with the noise impact it creates [25]. Generally speaking, a single rapid discharge of non-contact CMSs lasts for about 20 min [11] and requires a continuous supply of about 1uW UV light [26]. To ensure precise discharge, the light output performances of light sources must be improved. Considering the influence of UV light transmission and surface absorption, the steady-state performance of the light output needs to meet the stability requirement of 0.1%/h.

In the past, traditional mercury lamps were commonly used as light sources for UV applications, including the CMS carried by the LISA Pathfinder, due to technological constraints [5]. However, mercury lamps have been criticized for their shortcomings, such as slow turn-on response, limited dynamic range, sensitivity to temperature changes, limited lifespan, radio frequency interference, and electromagnetic interference [27]. Fortunately, in recent years, AlGaN-based UV LEDs with peak wavelengths below 270 nm have become available, providing significant advantages in energy consumption, package, stability, service lifespan, and high-frequency modulation performance. Consequently, UV LEDs have emerged as powerful alternatives to mercury lamps [9]. However, AlGaN-based UV LEDs are still in the exploratory stage with generally low luminous efficiency [28] and require long-term operation at low power in CMSs, making accurate modeling and control crucial for related application system designs.

The AlGaN-based UV LED used in this paper was UVTOP250-HL-TO39. According to Figure 1, this device has a peak wavelength of 255 nm and a half-wave width of 11nm. It is rated for a power output of 0.3 mW at 20 mA and has a thermal resistance of 50 °C/W, while the rated luminous efficiency is 2.5‰. This device is packaged in TO-39 and has a hemispherical sapphire lens, which results in a narrow beam angle of only $7^{\circ }$.

Because of the extremely fast response speed and MHz modulation ability of UV LEDs, the time delay in their model is almost negligible. Therefore, it is common to obtain their linear model by calibrating the relationship between current and optical power [29], which can lead to a better controller for applications with low accuracy requirements. However, it is worth noting that UV LEDs are nonlinear systems that integrate optical, electrical, and thermal characteristics. Although thermal characteristics do not explicitly participate in the electroluminescence process, their influence on luminous efficiency is very important [30]. Unfortunately, little attention has been paid to this issue in the past.

## 3. Low-Power PET Model and Fuzzy Adaptive PID Controller Design

### 3.1. Low-Power PET Model of UV LEDs

Since the core of a UV LED is a P-N junction [31], its V-I characteristics can be approximately expressed by the Shockley equation: (1)$$i=I_{s}\left(e^{\frac{qu}{nkT}}-1\right)$$
where *i* is the forward current, *u* is the forward voltage, $I_{s}$ is the reverse saturation current, *q* is the amount of electronic charge, *k* is the Boltzmann constant, *T* is the thermodynamic temperature, and *n* is a constant (ideal factor). From Equation (Equation 1), the relationship between *u* and *i* can be obtained, namely: (2)$$u=\frac{nkT}{q}\left[ln\left(i+I_{s}\right)-lnI_{s}\right]$$

UV LEDs are very sensitive to voltage changes, so they are usually driven by currents. The relationship between electrical power *P* and *i* can be obtained from Equation (Equation 2). However, since $I_{s}$ is very small, *P* can be approximately expressed as follows: (3)$$P=ui\approx \frac{nkTi}{q}\left[lni-lnI_{s}\right]$$

When *i* is small (i.e., low power operation in CMSs), the logarithmic term of *i* in Equation (Equation 3) can be expanded by the Taylor series, and the approximate expression of the first two terms can be obtained as follows: (4)$$P=\frac{knTi}{q}\left[\sum_{m=1}^{\infty }\;{\left(-1\right)}^{m-1}\frac{{\left(i-1\right)}^{m}}{m}-lnI_{s}\right]\approx \frac{knT}{q}\left[-\frac{i^{3}}{2}+2i^{2}-\left(\frac{3}{2}+lnI_{s}\right)i\right]$$

Among them, the third and second terms of *i* are much smaller than the first term, so *P* can be further approximated as a linear function of *i*, and a small constant *b* is added for correction; that is: (5)$$P=\frac{-knT}{q}\left(\frac{3}{2}+lnI_{s}\right)i+b\equiv ai+b$$

Assuming that $P_{v}$ is the optical power, it has the following relationship with *P*: (6)$$P_{v}=E_{v}P$$
where $E_{v}$ is the luminous efficiency that represents the ratio of electrical power converted into light output. Previous studies have shown that it is mainly affected by thermal characteristics and will decrease linearly with a constant coefficient $k_{e}$ as the junction temperature $T_{j}$ increases [17,18], namely: (7)$$E_{v}=k_{e}T_{j}\;\left(k_{e}<0\right)$$

Since UV LEDs are packaged with TO-39, $T_{j}$ is not easy to measure, so $E_{v}$ cannot be obtained directly by Equation (Equation 7). As the case temperature $T_{c}$ will be monitored by temperature sensors in CMSs, $T_{j}$ can be replaced by measurable $T_{c}$.

In the actual working process of UV LEDs, a large part of *P* is used for heating in addition to luminescence, resulting in an increase of $T_{j}$. The relationship between the thermal power $P_{h}$ and *P* is as follows: (8)$$P_{h}=E_{h}P$$
where $E_{h}$ is the thermal efficiency, and $E_{h}=1-E_{v}$ if other small losses are ignored.

To maintain the proper operating temperature of UV LEDs, it is necessary to use a heat sink to enclose the device in an actual CMS, as shown in Figure 2a. Moreover, to facilitate the analysis, we abstracted a simplified model from this real thermal structure as shown in Figure 2b, where $T_{a}$ is the ambient temperature, $R_{jc}$ and $R_{hs}$, respectively, represent the junction-to-case thermal resistance and the thermal resistance of the heat sink, while other thermal conductors are ignored due to their small thermal resistance.

Thus, $T_{j}$ can be expressed as follows: (9)$$T_{j}=T_{c}+R_{jc}P_{h}$$

We bring Equations (Equation 9) into Equation (Equation 7): (10)$$E_{v}=k_{e}T_{c}+k_{e}R_{jc}P_{h}$$

We combine Equations (Equation 6), (Equation 8), and (Equation 10): (11)$$P_{v}=k_{e}T_{c}P+k_{e}R_{jc}E_{h}P^{2}$$

Since CMSs only require uW-level output, the electrical power of UV LEDs will be in the order of mW, so the quadratic term of *P* in Equation (Equation 11) is much smaller than the first term, and it can be simplified as follows: (12)$$P_{v}=\left(k_{e}T_{c}+c\right)\left(ai+b\right)+d$$
where *c* is a constant to correct the error caused by neglecting the quadratic term in Equation (Equation 11). In addition, due to the difference between the calibration operating point and the actual operating point, a certain constant error may be introduced; thus, a constant correction term *d* may be added to improve the accuracy of modeling.

Equation (Equation 12) provides an approximate relationship between optical power and two measurable independent variables, namely the driving current and case temperature. Moreover, only two linear curves are needed for modeling in practical applications, which is very convenient for UV LEDs to be calibrated periodically to correct the accurate discharge without interrupting the operation of CMSs, based only on the data collected by the CMS in real time. Moreover, when an independent variable is fixed (typically the case temperature is kept constant), it becomes a traditional linear model.

### 3.2. Design of Fuzzy Adaptive PID Controller

Effective optical power control of UV LEDs is essential for stable and reliable charge management, especially in the presence of uncertainties caused by thermal effects and attenuation effects, to ensure accurate discharge and low noise levels [32]. Moreover, fuzzy adaptive PID control can meet these requirements well. However, the dynamic performance and steady-state performance of a control system are essentially irreconcilable trade-offs [33]. To strike a balance between them, we propose an optical power control system using the FA_PID algorithm with a switch, which is shown in Figure 3.

This system uses a two-dimensional fuzzy controller, which converts the optical power control error *e* and error rate ec into fuzzy quantities and generates the fuzzy quantities of controller parameter changes suitable for the current system state adjustment in the inference machine based on the rules in the rule base. Then, these fuzzy quantities are transformed into accurate quantities through the defuzzifier by the centroid method. The quantization factors, $e_{P}$, $e_{I}$, and $e_{D}$, are then used to quantify them into ΔP, ΔI, and ΔD, respectively, which are sent to the PID controller. By combining these values with the initial parameters $P_{0}$, $I_{0}$, and $D_{0}$, the self-tuning of the PID controller is achieved.

The addition of a fuzzy controller is identified as the main factor responsible for reducing the dynamic performance. To balance the dynamic performance and steady-state performance, a switch with threshold $e_{0}$ is introduced to coordinate the working state of the controller at different stages. During the stable stage ($\left|e\right|\leq e_{0}$), the fuzzy controller sends its output to the PID controller through the switch, significantly improving the steady-state performance. In the rising stage ($|e|>e_{0}$), the switch disconnects the fuzzy controller, and the PID controller generates a strong control output to meet the dynamic performance requirements. Moreover, by adjusting $e_{0}$, the proposed system can achieve comprehensive optimization of the dynamic performance and steady-state performance. Therefore, the above controller can be described as follows: (13)$$\Delta u\left(k\right)=\left\{\begin{array}{c} P\left[e\left(k\right)-e\left(k-1\right)\right]+Ie\left(k\right)+D\left[e\left(k\right)-2e\left(k-1\right)+e\left(k-2\right)\right]\;\;\;\;\;\left(e\leq e_{0}\right)\\ P_{0}\left[e\left(k\right)-e\left(k-1\right)\right]+I_{0}e\left(k\right)+D_{0}\left[e\left(k\right)-2e\left(k-1\right)+e\left(k-1\right)\right]\;\;\;\;\;\left(e>e_{0}\right) \end{array}\right.$$
where $P=P_{0}+\Delta P$, $I=I_{0}+\Delta I$, and $D=D_{0}+\Delta D$.

The membership function design is crucial for the above controller design, while the sharpness of the membership function determines the resolution and sensitivity of the controller [34]. In order to achieve better control, it is recommended to use low-resolution fuzzy sets in areas with larger errors and high-resolution fuzzy sets in areas with smaller errors. Based on the balance between control effect and computational complexity, we divided all input and output variables into seven fuzzy subsets ([NB, NM, NS, ZO, PS, PM, PB]), and used the same membership function shown in Figure 4.

The quality of a fuzzy controller depends largely on the quality of the control rules. Based on the operational experience of CMSs and the existing PID controller parameter tuning experience [35,36], the following fuzzy control rules have been established:When *e* is large, in order to eliminate the error as soon as possible and prevent the differential supersaturation that may be caused by the excessive moment of *e*, ΔP takes a large value, and ΔI and ΔD take a small value or zero.When *e* is small, in order to further eliminate the error and prevent the oscillation caused by excessive overshoot, ΔP should be reduced, ΔI should be small, and ΔD should be moderate to ensure the response speed of the system.When *e* is very small, in order to eliminate the static error and avoid oscillation near the set value, ΔP continues to decrease, ΔI remains unchanged or slightly larger, and ΔD can be slightly larger.The magnitude of ec indicates the rate of error changes. The larger the ec, the smaller the ΔP, the larger the ΔI, and vice versa.When *e* and ec have the same sign, the controlled variable deviates from the given value direction, and the control action should be strengthened to make the error change in the direction of reduction. Thus, a larger ΔD and a smaller ΔI should be taken, and ΔD should not be too large.When *e* and ec have different signs, the controlled variable changes in the direction close to the given value, so when *e* is large, take a smaller ΔP or zero to accelerate the dynamic process.

Based on the above rules and fuzzy set division, a rule base consisting of 49 rules was designed (presented in Table 1, Table 2 and Table 3) by using the conditional statements “if *E* and EC then ΔP, ΔI, ΔD”.

The input–output relationship surfaces of the fuzzy controller designed by the above steps are shown in Figure 5. The different colors reflect the magnitude of the values, with warmer colors indicating higher values and cooler colors indicating lower values.

## 4. Results and Discussion

The experimental setup is depicted in Figure 6. A programmable power supply (E36312A) was used to drive the UV LED, whose UV light was directed vertically onto the sensitive area of the photodiode probe (818-UV/DB) through an aluminum coupling structure. The optical power data were then converted into digital signals by an optical power meter (844-PE-USB). The case temperature of the UV LED was monitored by a thermal resistance sensor (PT-1000) that was affixed to the case and converted into digital signals through a data acquisition system (DAQ970A). To facilitate experiment operations and data acquisition, a special LabVIEW automatic software was compiled.

### 4.1. Modeling Low-Power PET Characteristics of UV LEDs

In order to construct the L_PET model of UV LEDs, two linear curves are needed. When calibrating the current-electric power curve, the case temperature is kept constant at 25 °C, while the driving current changes between 0 and 7 mA. Moreover, the terminal voltage data under each driving current are recorded to calculate the electric power. As described in Equation (Equation 5), a linear curve, as shown in Figure 7a, can be obtained. It is worth noting that in the practical application of charge management, even considering the coupling and transmission loss, only dozens of uW UV lights are needed, so the driving current will not exceed the calibration range here. To calibrate the case temperature–luminous efficiency curve, a constant current of 3.5 mA is maintained while recording the optical power and terminal voltage data as the case temperature changes. As the case temperature is kept constant at 25 °C in CMSs, the operating characteristics from 25 °C to 26 °C are our focus. By calculating the ratio of optical power to electrical power at each temperature, as described in Equation (Equation 10), a linear curve, as shown in Figure 7b, can be obtained. Obviously, the curve is not well fitted in the middle section, which may be due to environmental factors such as airflow affecting the accuracy of temperature monitoring. However, these effects generally remain within an acceptable range.

As described in Equation (Equation 12), the L_PET model can be established by calibrating the linear curves shown above. In addition, a traditional current-optical power linear fitting model was also established based on the same data for comparison, which is shown in Figure 8. Obviously, when the UV LED is operated at low power, there is a noticeable nonlinearity exhibited between the current and optical power, making the traditional model that describes this relationship only by linear fitting, without considering the thermal characteristics, not very good in practice, especially at the high current end, which explains the poor prediction of the traditional linear model.

To compare the modeling effectiveness, two experiments were conducted for the L_PET model and the traditional model. In the first experiment (steady-state test), the driving current remained constant while the case temperature increased due to heating. Real-time data were collected for predictions using the L_PET and traditional models, and these prediction results were then compared with the actual optical power data, as shown in Figure 9. The second experiment (dynamic test) involved continuous step changes in the driving current, and the same comparison results are displayed in Figure 10. It is important to note that due to the difference between the calibration operating point and the actual operating point, the original prediction results will be poor due to the existence of constant errors. Therefore, before each group of tests, we first conducted a pre-test for about 10 s; the average error was used as a constant correction term to improve the prediction accuracy for both models.

Both tests lasted about 150 s, and Table 4 presents the analysis, which clearly indicates that the L_PET model is more effective when UV LEDs operate stably, with a maximum prediction error of only 14.4 nW. As time passes or the device’s thermal characteristics gradually emerge, the advantage of the L_PET model becomes more apparent, and its prediction is significantly better than that of the traditional model, without considering the thermal characteristics. However, the dynamic test results of the L_PET model were unsatisfactory, with poor prediction performance before and after the sudden change in step current. Moreover, the thermal capacity of UV LEDs could be responsible for this outcome, which caused a non-steady-state thermal model, leading to significant errors in the sudden current change. Nevertheless, the worst-case relative error remains in the order of a thousandth, and the overall performance is slightly better than that of the traditional model, which still has potential for application.

### 4.2. Results of the Optical Power Control System for UV LEDs

To develop a controller that meets the requirements of CMSs, a control simulation model was built in MATLAB/Simulink, as depicted in Figure 11. The FA_PID controller with a switch and the established L_PET model were both implemented as sub-modules. Due to the adoption of thermal management, the case temperature of the UV LED can be considered constant at 25 °C. Moreover, the current source was approximated as a first-order inertia element with parameters obtained through special calibration experiments.

After conducting numerous simulations, we obtained optimal controller parameters, as shown in Table 5.

To evaluate the effectiveness of the FA_PID controller with a switch, an optical power control experiment was conducted in the experimental setup, as shown in Figure 6. The FA_PID controller and an independent PID controller for comparison were implemented using LabVIEW software.

In this experiment, the set value of the controller was set to 1 uW to simulate the long-term working condition of UV LEDs in CMSs. Figure 12a shows the rising processes of the two controllers. The analysis indicates that the FA_PID controller demonstrates aggressive initial parameters in its pursuit of dynamic performance, resulting in a faster response speed than the PID controller during the first 90% of the rising process. However, during the following 10%, its speed decreased rapidly due to the addition of a fuzzy controller that prioritizes steady-state performance. Despite this dip in speed, the overall response speed still had an advantage. The integral absolute errors (IAE) for both PID and FA_PID control results were 5.8452 uW and 4.1853 uW, respectively, within 100 s of this experiment.

Compared to dynamic performance, CMSs place more emphasis on steady-state performance. Therefore, following the aforementioned experiment, we continuously monitored the fluctuation of the optical output for 30 min. The results (presented in Figure 12b and Table 6) indicate that the FA_PID controller has a more pronounced effect on suppressing fluctuations than the traditional PID controller, and it can reduce the fluctuation range of the optical output to about 0.67 nW. Furthermore, considering the fact that a single discharge in CMSs is completed within 30 min, the steady-state performance already meets the stability requirement of 0.1%/h.

By analyzing the dynamic performance and steady-state performance, the advantages of the FA_PID controller with a switch over the PID controller are well highlighted, and it achieves better performance in both aspects, which is undoubtedly helpful for the use of UV LEDs and the operation of the whole CMS.

## 5. Conclusions

UV LEDs play a crucial role in the construction of non-contact CMSs for inertial sensors, and their light output performances directly affect the effectiveness of charge management. This paper focuses on modeling and control strategies to improve the light output performances of UV LEDs operating at low power. In terms of modeling, a low-power PET model is proposed that comprehensively considers the optical, electrical, and thermal characteristics of UV LEDs, providing a more comprehensive understanding of the operating characteristics of UV LEDs while being practical for applications. Test results show that the proposed model has an average prediction error of only 5.8 nW when the light source output is stable, making it suitable for designing and analyzing UV LED-related systems. Based on this model, an optical power control system utilizing a fuzzy adaptive PID controller was developed, which includes a switch to adjust the self-tuning function of the fuzzy system and achieve a balance between dynamic and steady-state performance requirements according to different response stages. Test results demonstrate that the proposed controller effectively reduces the UV LED light output range to only 0.67 nW during a single discharge task, meeting the charge management requirements of high-precision inertial sensors for space gravitational wave detection.

Overall, this paper provides valuable insights into the development of UV LED-related systems. The proposed model and control strategy offer significant improvements in the light output performances of UV LEDs in CMSs. However, it is worth noting that the presented model assumes the steady-state operation of UV LEDs. With the growing demand for versatile operation of non-contact discharge, the AC drive of UV LEDs is becoming increasingly interesting. Future improvements should focus on extending the model to address AC drive scenarios. Furthermore, our control work mainly focuses on individual UV LEDs, while other parts of the CMS, such as the driver circuit and the optical coupling interface, which are critical to the operating state of UV LEDs, are not considered. This suggests that future work should include system-level design considerations and explore how these components affect the output performances of UV LEDs.

## Figures and Tables

**Figure 1 sensors-23-05946-f001:**
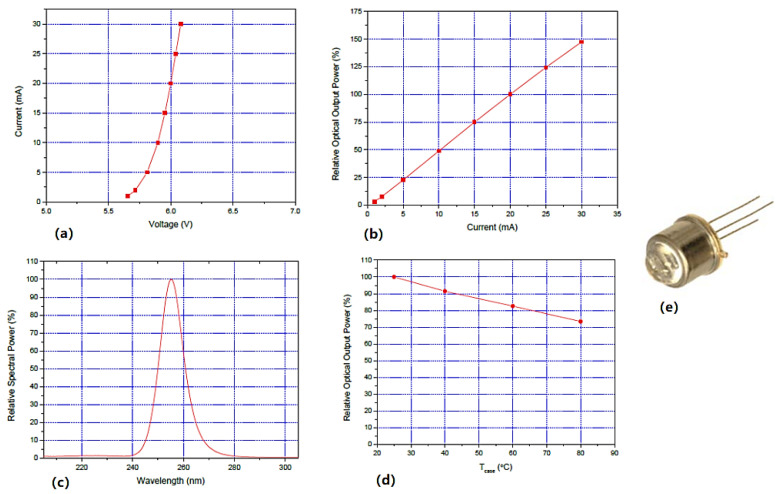
UVTOP250-HL-TO39 (**a**) current—voltage curve, (**b**) rel. optical power—current curve, (**c**) spectrum curve, (**d**) rel. optical power—case temperature curve and (**e**) photograph.

**Figure 2 sensors-23-05946-f002:**
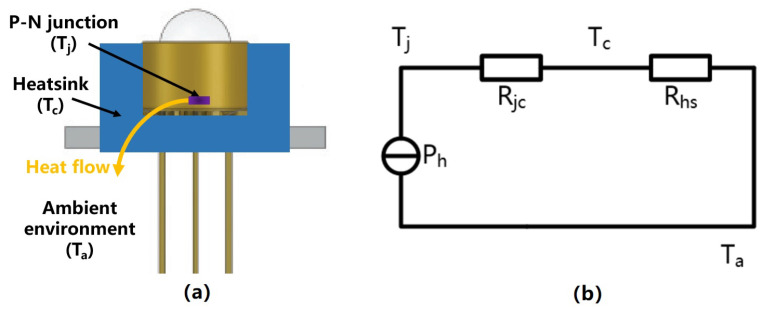
(**a**) Thermal structures of UV LEDs in CMSs. (**b**) Simplified steady-state thermal model.

**Figure 3 sensors-23-05946-f003:**
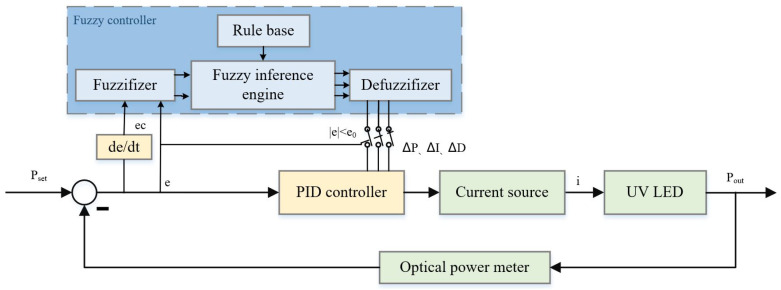
Schematic diagram of a UV LED optical power control system based on the FA_PID controller with a switch.

**Figure 4 sensors-23-05946-f004:**
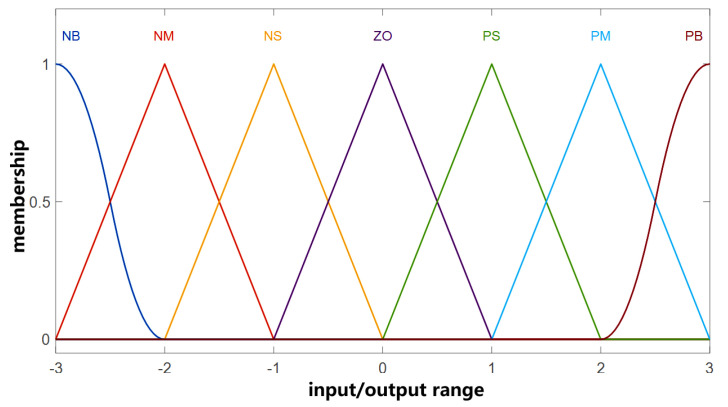
Membership functions of input/output variables for the fuzzy controller.

**Figure 5 sensors-23-05946-f005:**
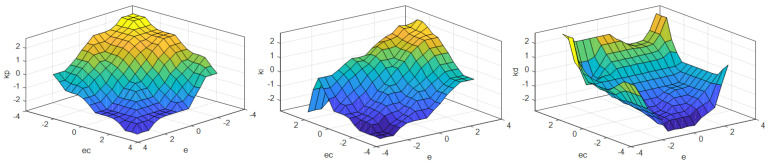
Input–output relationship surfaces of the fuzzy controller.

**Figure 6 sensors-23-05946-f006:**
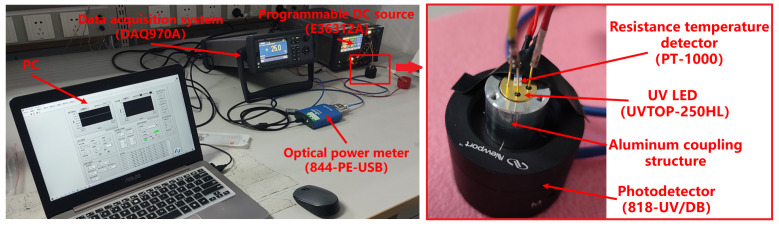
The experimental setup of this paper.

**Figure 7 sensors-23-05946-f007:**
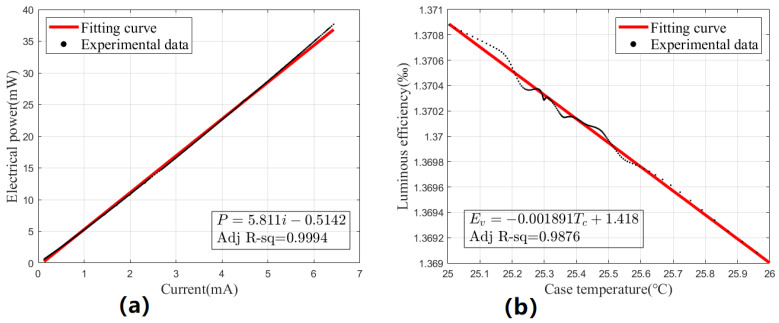
(**a**) Linear fitting results of current-electrical power. (**b**) Linear fitting results of case temperature–luminous efficiency.

**Figure 8 sensors-23-05946-f008:**
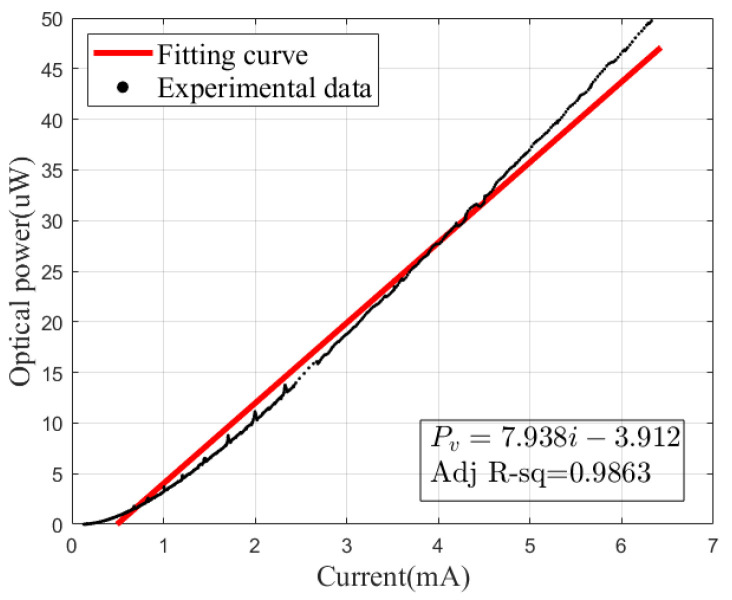
Traditional linear model for UV LEDs.

**Figure 9 sensors-23-05946-f009:**
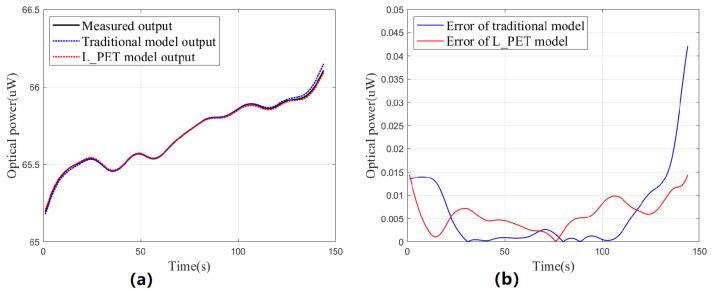
Results of the steady-state test. (**a**) Comparison of prediction results. (**b**) Comparison of prediction errors.

**Figure 10 sensors-23-05946-f010:**
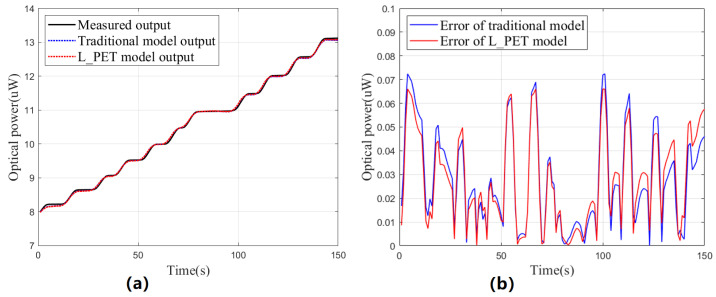
Results of the dynamic test. (**a**) Comparison of prediction results. (**b**) Comparison of prediction errors.

**Figure 11 sensors-23-05946-f011:**
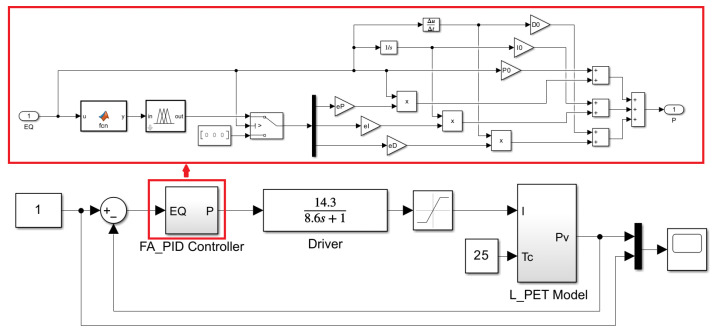
FA_PID optical power control simulation link based on L_PET model.

**Figure 12 sensors-23-05946-f012:**
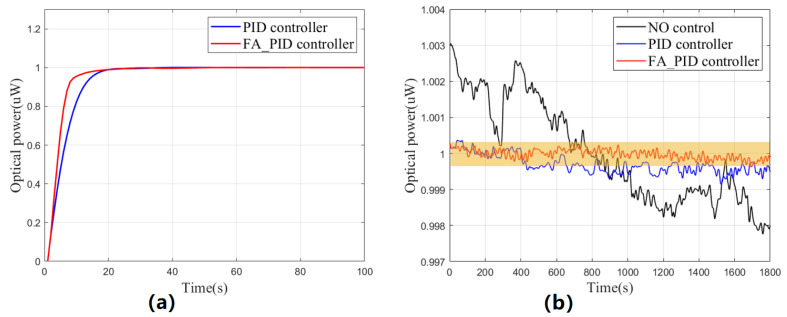
(**a**) Comparison of the rising process between FA_PID controller and PID controller. (**b**) Comparison of the optical output improvements between FA_PID controller and PID controller.

**Table 1 sensors-23-05946-t001:** Tuning rules for ΔP.

		E						
		NB	NM	NS	ZO	PS	PM	PB
**EC**	NB	PB	PB	PM	PM	PS	ZO	ZO
	NM	PB	PB	PM	PS	PS	ZO	NS
	NS	PM	PM	PM	PS	ZO	NS	NS
	ZO	PM	PM	PS	ZO	NS	NM	NM
	PS	PS	PS	ZO	NS	NS	NM	NM
	PM	PS	ZO	NS	NM	NM	NM	NB
	PB	ZO	ZO	NM	NM	NM	NB	NB

**Table 2 sensors-23-05946-t002:** Tuning rules for ΔI.

		E						
		NB	NM	NS	ZO	PS	PM	PB
**EC**	NB	NB	NB	NM	NM	NS	ZO	ZO
	NM	NB	NB	NM	NS	NS	ZO	ZO
	NS	NB	NM	NS	NS	ZO	PS	PS
	ZO	NM	NM	NS	ZO	PS	PM	PM
	PS	NM	NS	ZO	PS	PS	PM	PB
	PM	ZO	ZO	PS	PS	PM	PB	PB
	PB	NB	NB	NM	NM	NS	ZO	ZO

**Table 3 sensors-23-05946-t003:** Tuning rules for ΔD.

		E						
		NB	NM	NS	ZO	PS	PM	PB
**EC**	NB	PS	NS	NB	NB	NB	NM	PS
	NM	PS	NS	NB	NM	NM	NS	ZO
	NS	ZO	NS	NM	NM	NS	NS	ZO
	ZO	ZO	NS	NS	NS	NS	NS	ZO
	PS	ZO	ZO	ZO	ZO	ZO	ZO	ZO
	PM	PB	NS	PS	PS	PS	PS	PB
	PB	PB	PM	PM	PM	PS	PS	PB

**Table 4 sensors-23-05946-t004:** Prediction error comparison between the L_PET model and traditional model.

	Model	Maximum (nW)	Mean (nW)	SD (nW)	IAE (uW)
Steady-state test	Traditional	42.1062	5.9819	8.1184	0.8614
	L_PET	14.4455	5.8485	3.1558	0.8322
Dynamic test	Traditional	72.4263	27.8624	20.0623	4.1794
	L_PET	66.0562	27.7786	19.3699	4.1668

**Table 5 sensors-23-05946-t005:** Parameters used in the FA_PID Controller.

Parameter	Meaning	Value
$P_{0}$, $I_{0}$, $D_{0}$	Initial parameters of PID	125.087, 0.035, 0.007
$e_{P}$, $e_{I}$, $e_{D}$	Output scaling factor	1, 0.01, 10
$e_{0}$	Switching threshold	0.1

**Table 6 sensors-23-05946-t006:** Comparison of the suppression of optical output fluctuations between the FA_PID controller and PID controller.

Controller	Range (nW)	Mean (uW)	SD (nW)	IQR (nW)	IAE (uW)
NO control	5.2870	1.0000	1.4080	2.5486	2.2284
PID	1.2234	0.9997	0.2528	0.3113	0.6391
FA_PID	0.6731	1.0000	0.1287	0.1987	0.1957

## Data Availability

The data and the source code are publicly available at https://github.com/lengjugh/Modeling-and-Control-for-UV-LEDs (accessed on 27 May 2023).

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
