# Peer review of "Photo-Electro-Thermal Model and Fuzzy Adaptive PID Control for UV LEDs in Charge Management"

_sensors, 2023, doi:10.3390/s23135946_

Round 1

Reviewer 1 Report

-Deeper lit review needs to be conducted to highlight state of the art

-Explicit contributions to literature needs listing

-An overview of the Fuzzy Inference system needs describing

-Discussions could be expanded

-The conclusion needs a more reflective depth, as well as a pathway for further work

-More references need to be added in the Appendix

n/a

Reviewer 2 Report

The article is well organized; the description of the problem is adequate.

To improve this article, from my point of view, the authors should consider the following comments:

- To cite some references that clarify how the fuzzy rules were established (lines 181-183).

- Possibly using the analysis of the error integrals could help to better visualize the results shown in the figures 9.b and 10.b, 12.a andtable 4.
